# Are Reproductive Traits Related to Pollen Limitation in Plants? A Case Study from a Central European Meadow

**DOI:** 10.3390/plants9050640

**Published:** 2020-05-19

**Authors:** Michael Bartoš, Štěpán Janeček, Petra Janečková, Eliška Chmelová, Robert Tropek, Lars Götzenberger, Yannick Klomberg, Jana Jersáková

**Affiliations:** 1Institute of Botany, The Czech Academy of Sciences, 37981 Třeboň, Czech Republic; lars.gotzenberger@ibot.cas.cz; 2Department of Ecology, Faculty of Science, Charles University, 12843 Praha, Czech Republic; janecek.stepan@centrum.cz (Š.J.); janeckova.petra@centrum.cz (P.J.); eliska.chmelova@entu.cas.cz (E.C.); robert.tropek@natur.cuni.cz (R.T.); klombery@natur.cuni.cz (Y.K.); 3Faculty of Science, University of South Bohemia, 37005 České Budějovice, Czech Republic; jersa@prf.jcu.cz; 4Biology Centre, Institute of Entomology, The Czech Academy of Sciences, 37005 České Budějovice, Czech Republic

**Keywords:** pollen limitation, supplemental hand-pollination, seed number, seed weight, floral traits, wet meadow

## Abstract

The deficiency of pollen grains for ovule fertilization can be the main factor limiting plant reproduction and fitness. Because of the ongoing global changes, such as biodiversity loss and landscape fragmentation, a better knowledge of the prevalence and predictability of pollen limitation is challenging within current ecological research. In our study we used pollen supplementation to evaluate pollen limitation (at the level of seed number and weight) in 22 plant species growing in a wet semi-natural meadow. We investigated the correlation between the pollen limitation index (PL) and floral traits associated with plant reproduction or pollinator foraging behavior. We recorded significant pollen limitation for approximately 41% of species (9 out of 22 surveyed). Seven species had a significant positive response in seed production and two species increased in seed weight after pollen supplementation. Considering traits, PL significantly decreased with the number of pollinator functional groups. The relationship of PL with other examined traits was not supported by our results. The causes of pollen limitation may vary among species with regard to (1) different reproductive strategies and life history, and/or (2) temporary changes in influence of biotic and abiotic factors at a site.

## 1. Introduction

Pollen limitation (i.e., limitation of seed production by deposition of pollen grains) is among the key factors affecting the fitness of individual plants and consequently, population dynamics and species survival [1]. Therefore, with the global pollination crisis [2,3], pollen limitation has become a key topic of ecology and conservation of plant communities [4,5,6]. Despite several decades of research, there is still no consensus on how widespread pollen limitation is in plant communities. The optimality theory [7] and sexual selection theory [8], but see [9] predict that pollen limitation should be rare. However, numerous empirical studies showed pollen limitation as a relatively common phenomenon [10,11]. A review of 306 plant species found evidence of pollen limitation (within an individual site) in 73% of the studies [12]. Consequently, this suggested insufficient pollen receipt to be the major cause of reduced fruit production [12]. Nevertheless, the existing geographical bias of available detailed data [13] limit any strong generalizations on the extent of pollen limitation, as well as causes and consequences in individual plant species and in communities.

Pollen supplementation experiments represent a standard method for pollen limitation quantification [14]. Based on saturation by manually applied additional pollen to flowers, it allows a robust subsequent comparison of their fruit sets and/or seed sets with naturally pollinated flowers [14]. Besides the effects of pollen saturation on the quantitative characteristics, possible trade-offs in resource allocation can be evaluated also by a qualitative comparison of seed or fruit sets (e.g., by their size or weight, [15]). Nevertheless, published results from pollen supplementation experiments are predominantly based on single-species case studies. Therefore, they may not be representative of the realized pollen limitation in communities [16]. Plant species, as well as individuals in the population, may not be equally sensitive to changes in environmental and associated biotic conditions because the possible lack of pollination depends on the ecological context, plant life history, and type of breeding system [1].

The shift of plant species to outcrossing can be caused by specific plant trait evolution regardless of the possible consequence of pollen limitation [12]. However, the correlation of pollen limitation with various life-history and ecological traits was tested in only a few comparative studies [11,17]. In 224 species from 64 families of flowering plants, Larson and Barrett [11] revealed pollen limitation as less intense in species which are self-compatible, autogamous, monocarpic, herbaceous, nectariferous, and occurring in open habitats and temperate regions. Although self-incompatible plants are generally expected to be more pollen limited than self-compatible plants [10], this assumption may not always be true. As discussed by García-Camacho and Totland [17], self-compatible species may potentially receive more compatible pollen on their stigmas than self-incompatible species, but resource limitation might not let them take advantage of it. Thus, constraints from specific abiotic conditions could theoretically explain the similarity between self-compatible and self-incompatible species [17]. Furthermore, comparisons of pollen limitation between phenotypically specialized and generalized flowers reported ambiguous results. Larson and Barrett [11] found that species with specialized floral morphology and less accessible nectar did not differ from those with generalized morphology in the level of pollen limitation. Contrarily, Lázaro et al. [18] recorded that species with specialized flowers were more pollen limited than those with generalized flowers. Therefore, individual floral traits can explain only a small part of variation in pollen limitation [11].

New insights into the variation of pollen limitation causes could be provided by exploration of correlative effects between multiple reproductive and functional traits and pollen. For example, even key traits like dichogamy or clonality have not been thoroughly explored in this context. While dichogamy level has been suggested as ensuring higher autonomous seed set in plants exposed to outcross pollen limitation [19], clonality may provide reproductive advantage for obligate outcrossing species that are in the higher risk of pollen limitation [20].

In this study, we applied pollen supplementation to evaluate the level of pollen limitation in a community of flowering plant species in a wet meadow in a fragmented cultural landscape in Central Europe. Consequently, we correlated the pollen limitation with multiple functional traits of the plant species. We hypothesized that the degree of pollen limitation of plant species will be influenced by (i) a type of breeding system, (ii) floral traits important for pollinator attraction and foraging technique, and (iii) their degree of functional specialization on pollinators. We expected that plants visited by a broad spectrum of different insect functional groups (i.e., bees, flies, beetles, etc.) will be less pollen limited. We also predicted that the lower pollen limitation would occur in pollinator-attractive plants with abundant nectar rewards and/or more open flowers. Last but not the least, we provided a comprehensive pollen limitation dataset from Central Europe, a region previously largely neglected in pollination networks and pollen limitation studies [13].

## 2. Results

### 2.1. Seed Production and Seed Weight

The differences in both seed production and seed weight between pollen-supplemented and naturally pollinated flowers at the community level were statistically significant (permutational MANOVA; Pseudo-F = 3.99, *p* = 0.023, and Pseudo-F = 6.92, *p* = 0.005, respectively). At the species level, we found a statistically significant positive increase in seed production after pollen supplementation in seven species (i.e., Table 1): *Anemone nemorosa*, *Lysimachia vulgaris*, *Lychnis flos-cuculi*, *Potentilla palustris*, *Aegopodium podagraria*, *Ranunculus auricomus*, and *Stellaria graminea*. The mean weight per seed of the pollen-supplemented flowers was significantly higher in two species, *Lychnis flos-cuculi* and *Cardamine pratensis*. The capsula weight after the pollen supplementation significantly increased in *Dactylorhiza majalis*. PLs for all individual species are presented in Figure 1.

### 2.2. Traits Correlations

Our tests revealed that PL was significantly related only to the number of pollinator functional groups (Table 2 and Figure 2). We found no significant relationship between PL and other tested traits, including the multiple regression with all traits (F = 0.83, *p* = 0.57). The only trait selected by the AIC-based stepwise selection was again the plant specialization. All correlation indices between particular quantitative floral traits are presented (Appendix A).

Both models, the unimodal and the linear, were significant (unimodal: F = 7.95, *p* = 0.003, Figure 2B; linear: F = 4.39, *p* = 0.049, Figure 2A). However, because of the relatively small number of target plant species, this unimodal relationship may be greatly affected by outlying values at the edges.

## 3. Discussion

Pollen limitation is generally considered a common phenomenon and many comparative studies report relatively high occurrence (62–73%) in various habitats [1,10]. However, we only recorded significant pollen limitation for approximately 41% of species (9 out of 22 surveyed) in our wet meadow community. Our findings are in concordance with a similar unusually low occurrence of pollen limitation in a temperate grassland community in western Norway [21]. This study focused on pollen limitation and its relationship to plant species visitation rates and specialization levels and revealed only two out of eleven (~18%) studied plant species to be significantly pollen limited. Moreover, Bennett et al. [13] even documented no pollen limitation in investigated study of nine species in a Romanian meadow community. It might seem that the low levels of pollen limitation revealed in the pollen supplementation experiments are in agreement with the assumptions from the model by Haig and Westoby [7], which stipulates that seed set in flowering plants should be equally limited by both pollen and resource availability. It further suggests that pollen supplementation should not increase seed set in populations at their evolutionary equilibrium, because resources should be unavailable for maturation of their additional fertilized ovules. However, Burd [22] adjusted this model for stochastic variation in both ovule fertilization and resource availability, which made the model broadly in accordance with the recent meta-analysis [1,10], in which pollen limitation is found in most surveyed species.

The reported inconsistencies in the magnitude of pollen limitation could stem from several non-mutually exclusive reasons:(1)*Effect of sampling size and experimental design*. Using power tests (via simulation) for pollen supplementation experiments, Thomson [23] illustrated that moderate pollination deficits of up to 15% will usually not be detected with sample sizes of 20 individuals, and even 40 are insufficient for minor deficits. But, unfortunately, lower sampling effort (such as 20–30 individuals in our study) is an inevitable result of various logistic constrains and trade-offs between the data quantity and quality in most community studies [13,18,21,24].(2)*Publication bias*. The community approach, where multiple plant species are studied simultaneously, may lead to a better understanding of patterns in pollen limitation. It is because environmental characteristics, such as nutrient levels within a given community, are relatively homogenous in such studies and the role of plant traits in pollen limitation can, therefore, be better assessed. Nevertheless, there have been few studies focused on the relationships between plant traits and pollen limitation across whole communities [18,21,24,25,26]. All these studies recorded lower levels of pollen limitation in natural systems compared to the pollen limitation documented in comprehensive reviews that are mostly based on single-species studies [1,10]. Therefore, the publication bias, favoring statistically significant responses which then become available for further studies, together with the omission of “grey literature” and studies not written in English [27], complicates our understanding of pollen limitation [14].(3)*Effect of pollinator abundance*. Hegland and Totland [21] discussed their results of low pollen limitation in the context of a possible higher pollinator abundance in the studied community, which could substantially reduce the quantitative pollen limitation. A partial cause of low pollen limitation in our study could be that the targeted semi-natural locality is situated in a relatively well-preserved and mosaic-like landscape with a limited influence of intensive agriculture. Such semi-natural, diverse, and heterogeneous environments support pollination services [28,29] and thus increase the plant reproductive success, as suggested by Bennett et al. [13] in their Romanian meadow community.(4)*Effect of plant community composition and study species selection*. In our investigated community, only a few plant species with morphologically highly specialized flowers, which are expected to be more prone to pollen limitation, were present. Therefore, this community may have a lower pollen limitation than communities with a greater proportion of specialized flowers.(5)*Choice of the pollen limitation measure*. An important factor determining the recorded magnitude of pollen limitation may also be the choice of its measure. Knight et al. [14] compared 263 studies working with different measurements of the production component of reproduction and revealed the largest effect for relative fruit set, and the lowest effect for production of seeds/flower and seeds/fruit. However, because the magnitude of pollen limitation was inter-correlated among these response variables, Knight et al. [14] assumed that pollen limitation occurs simultaneously at different stages of the plant reproduction, but with varying intensity. Also in our study the numbers of pollen-limited species varied substantially between the two applied measures, seed production and seed mass. Furthermore, Hegland and Totland [21] pointed out that the two main components of plant reproductive success, seed production and seed mass, are often not included in the same studies.

In our studied community, two species showed significant positive seed weight response after pollen supplementation, though we expected the negative relationship. Several studies demonstrated that seed mass decreases with pollen availability because of seed size–number trade-off [1]. For example, Ågren et al. [30] recorded reduced mean seed size in hand-supplemented *Primula farinosa* by about 12%, but a larger total mass of seeds than in naturally pollinated plants. The opposite effect, i.e., increased seed weight after pollen supplementation could be explained by the increased pollen quality [21]. Aizen and Harder [31] suggested that the cross-pollen used for supplementation may have higher quality than the mixture of self- and cross-pollen available under the natural pollination. It seems evident that the magnitude of pollen limitation is dependent on the treatment level, e.g., whether the experimental design is applied only on a fraction of the plant’s flowers or on the whole plant [32]. Unfortunately, because we treated only flower pairs, we can only speculate on the proportion of resource allocation in our study [33]. The low differences in the seed weight between the treatments could be caused by the ability of the plant to compensate for any possible higher cost of an additional seed production induced by the supplemental pollination in only one flower.

Species with high PL values have specific ecological features. *Lysimachia vulgaris* is pollinated by highly specialized oil-collecting *Macropis* bees [34]. The species is also dominant in the locality, producing many flowers at same time, which suffer from competition for pollinators. An orchid *Dactylorhiza majalis* offers no reward to pollinators, which are deceived by showy flowers. Deceptive orchids typically produce little fruits [35]. *Anemone nemorosa* is flowering very early in the vegetation season when visitors are limited by unexpected weather conditions, especially by low temperature [36]. Species with low PL mainly belong to a group of plants with many, generalized flowers with easily available nectar rewards. But nectar is not the only reward which is offered by plants. Pollen is also a very important attractant for visitors. Unfortunately, we do not have adequate data about pollen production for the investigated plant species. However, the main pollinator group recorded on the studied locality appears to be hoverflies, feeding on both nectar and pollen. A wide range of hoverfly larvae are associated with accumulations of wet, rotting vegetation in ponds and ditches, which are common nearby. The other abundant groups were other flies and honey bees).

Despite analyses of several floral and life history traits connected to plant reproduction, we only found the significant relationship of PL to the number of pollinator functional groups. This finding is in accordance with the meta-analysis of pollen limitation in different world regions [16,37,38], where the more pollinator-specialized plant species were also more pollen limited (but see [21]). However, Lázaro et al. [18] pointed out that this relationship is not entirely clear and it is very important to distinguish between morphological (based on floral shape) and ecological (based on realized interactions) specialization. They found a strong negative relationship between pollen limitation and ecological generalization, but only for species with the morphologically specialized flowers. As a possible explanation they suggested that the morphologically specialized flowers benefit more from generalizing their pollination system in the lack of a primary pollinator [18]. The high ecological generalization may however result in the stronger pollen limitation because of lower flower-visitor diversity with abundant low-efficiency pollinators transporting high loads of incompatible pollen [39]. Accordingly, we recorded stronger pollen limitation in the species with specialized, as well as highly generalized pollination systems. This supports the prediction that many mutual relationships between plants and visitors should be non-linear [40,41].

However, assuming the validity of the linear model, we expect higher diversity in conspecific pollen load with increasing number of pollinator functional groups because of pollen grains coming from a wider range of donors. It could stimulate pollen competition and successful pollination. On the other hand, assuming the validity of the unimodal model which has much higher significance value than the linear model, increasing the number of pollinator functional groups may involve less-specific pollinators. These may clog stigmas with higher loads of heterospecific pollen, which could decrease the reproductive success of plants. Nevertheless, we can only speculate about the accuracy of using one model, because a detailed study on pollinator group effectiveness for each particular plant species would be necessary to make an unambiguous conclusion.

Our results describe pollen limitation based on observation from a single season. As seasonal course of climate affects both plant phenology and insect activity, thus we cannot exclude that pollen limitation will be different in other seasons.

## 4. Materials and Methods

### 4.1. Study Site

Our focal plant community was situated in a semi-natural wet meadow near the Chobotovský rybník pond in the landscape protected area of Železné hory (Bohemian-Moravian Highlands, Czech Republic; 535 m a.s.l., 49°46′57″ N, 15°50′17″ E). Mean annual temperature is 6.4 °C and annual precipitation is 745 mm. The subsoil is formed by fluvial sandy loam and sandy gravels. The meadow, with an area of 1.2 ha, is isolated from the surrounding agricultural landscape by a high forest. The meadow is mowed once a year. By phytosociological classification [42], the meadow belongs to the alliance *Calthion palustris* with vegetation dominated by *Agrostis canina*, *Scirpus sylvaticus*, *Lysimachia vulgaris*, and *Filipendula ulmaria*. We collected data on the 22 most abundant insect-pollinated plant species flowering between the beginning of May and mid-July 2017. During the vegetation season, 51 insect-pollinated species were flowering.

### 4.2. Pollen Limitation

Supplemental hand-pollination was applied on randomly selected individuals (20–30 per species) with at least two open flowers in similar phenological phases. One flower was supplemented by conspecific pollen from plants minimally two meters apart to reduce the genetic closeness and the second was left to natural pollination as a control. Both hand-pollinated and control flowers were marked by colored cotton yarn loosely knotted under the flowers. In species producing just a single flower (*Anemone nemorosa*) or compact inflorescences (*Bistorta major*) on a single shoot, we applied the treatments on two neighboring individuals. In Asteraceae species (*Crepis paludosa* and *Tephroseris crispa*) the treatments were applied on two whole capitula, and in Apiaceae species with compound umbels (*Aegopodium podagraria* and *Chaerophyllum aromaticum*) the treatments were applied on two umbellets from two different umbels. All species with flowers in compact and sequentially opening inflorescences (i.e., *Aegopodium podagraria*, *Bistorta major*, *Chaerophyllum aromaticum*, *Crepis paludosa* and *Tephroseris crispa*) were hand-pollinated repeatedly for several consecutive days throughout the whole flowering period. We collected anthers with visible pollen grains (verified by hand lens) and rubbed them over the receptive stigmas (successful deposition was again verified by hand lens). For compact inflorescences, capitulas, or umbels, we rubbed the whole donor unit over the recipient one. After the marked flowers wilted, their maturing ovaries were enclosed in fine nylon mesh bags to avoid any seed loss. We counted all viable seeds from the collected flowers or inflorescences and measured mean weight per seed. Because *Dactylorhiza majalis* produced numerous very small seeds, we used the capsule weight as a proxy for the number of developed seeds.

### 4.3. Plants Traits

We collated information on seven plant characteristics: Type of breeding system (i.e., extent of self-compatibility and autonomous selfing), level of dichogamy, clonality, amount of nectar reward, number of open flowers, and plant specialization on pollinator functional groups. Data on the extent of self-compatibility and autonomous selfing for 18 of the target species were obtained from our greenhouse pollination experiment [43]. Data on dichogamy were extracted from the Biolflor database [44] and transformed from the original seven categories to a continuous variable ranging from 0 to 0.5, with the value 0.5 denoting an absence of dichogamy (i.e., simultaneous presence of male and female organs). The missing data on breeding type (four cases) and level of dichogamy (two cases) were replaced by average values from the whole dataset. The clonal multiplication (i.e., number of vegetative offspring per maternal shoot per year) was extracted from the Clo-Pla database [45]. The daily sugar production in nectar reward was determined in the field for 15 flowers per plant species. Flowers from different plant specimens were bagged at their full anthesis for 24 h after which nectar was extracted. Nectar was washed with distilled water using a 100-µL Hamilton syringe and stored in a refrigerator prior to freezing, following Morrant et al. [46]. The amount of nectar sugars was quantified by high-performance liquid chromatography (HPLC) using the ICS-3000 system (Dionex), with an electrochemical detector and CarboPac PA 1 column. The nectar production was expressed in milligrams of nectar sugars per flower/day (Table 3). Mean number of open flowers per species was calculated from 60 specimens per species from three meadows in the study region.

Plant functional specialization was expressed as the number of pollinator functional groups that touched anthers and/or stigmas during foraging. The pollinator spectrum for each plant species was counted from videos recorded in the field using portable video systems of VIVOTEK (IB8367-T) and MILESIGHT (MS-C2962-FPB-IR60m) cameras. In total, 72 h (equally covering day and night) per plant species were recorded in three different localities in the vicinity of the study area. All pollinators were split into eleven functional groups: ants, beetles, bumblebees, butterflies, honeybees, hoverflies, long-tonged flies, moths, other bees, other flies, and other hymenopterans (Table 4). Groups which were represented by fewer than three visitors per plant species were excluded from the analyses to avoid random visits.

### 4.4. Data Analysis

Differences in reproductive success between supplemental hand-pollination and natural pollination at community and species levels were tested for both seed production and seed weight. Because the data contained many zero values and even after transformation did not meet the normality assumption, we applied non-parametric tests. At community level we used permutational MANOVA with permutation of residuals under a reduced model, where treatment served as fixed and plant species as random factor. At species level we used a one-sided test in non-parametric permutational ANOVA. Both tests were done within the PERMANOVA package in Primer 6 software [47].

We calculated the pollen limitation index (PL) as PL = (P_s_ − P_o_)/P_max_ (P_s_ or P_o_) [48], where P_s_ is the number of seeds from pollen-supplemented flowers, P_o_ is the number of seeds from open-pollinated flowers, and P_max_ is the larger of the two values (P_s_ or P_o_). For all subsequent analyses, similarly to Larson and Barrett [11], we established zero as the lower boundary of the PL, because any negative indices likely resulted from a potential experimental error [49], and therefore are not meaningful in the context of our study.

Although most studies assumed a linear relationship between possible plant seed set and traits, some studies predicted numerous relationships between plant and visitors to be non-linear [41,50]. Thus, all correlations of PL with plant characteristics were tested using both simple and multiple linear as well as unimodal regressions. Because of right-skewed distribution, the values for nectar production and number of flowers were log-transformed prior to analysis. For selection of the best model we used AIC stepwise selection. All analyses, unless otherwise specified, were conducted using R [51].

## 5. Conclusions

Our study recorded significant pollen limitation for approximately a third of species occurring in a wet meadow community. It was much lower than what has been reported in the previous reviews of single species studies, but higher when compared with all other community level studies. The discrepancy in the results of these studies can be attributed to several issues, such as sampling and publication biases. Except for the number of pollinator functional groups, we could not attribute pollen limitation to the other measured floral and life history traits. Therefore, some additional traits may also be contributing to patterns of pollen limitation. Such additional traits could be extrinsic traits (e.g., regional plant diversity) because interactions between extrinsic and floral or life history traits may be the major driver of pollen limitation in communities [52]. Finally, other overlooked and possibly important factors can be spatial and temporal variations in pollen limitation within and among communities.

## Figures and Tables

**Figure 1 plants-09-00640-f001:**
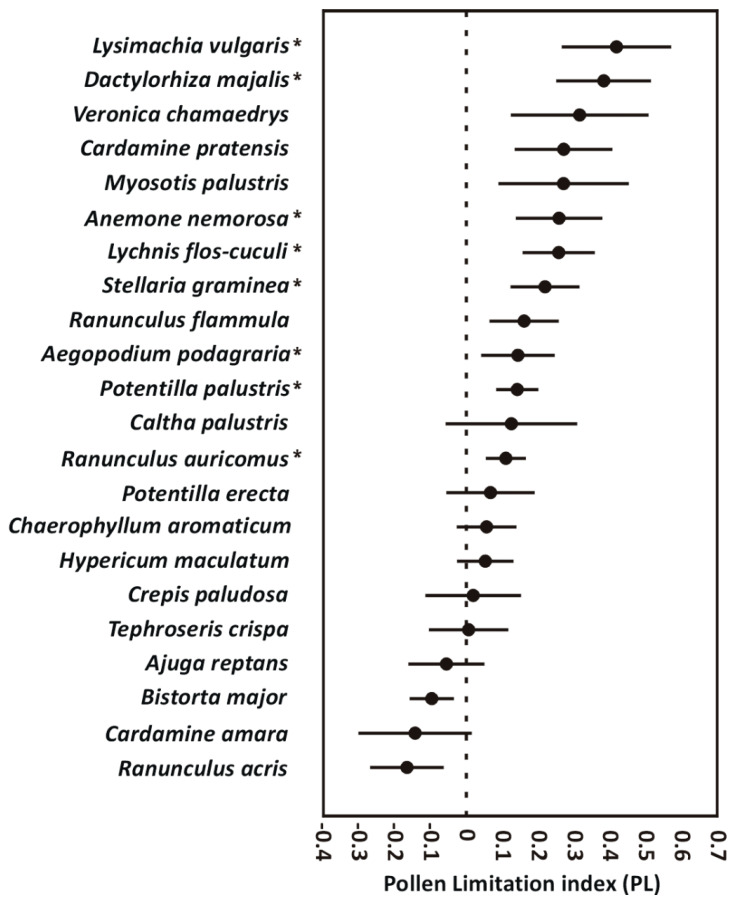
Pollen limitation index (PL) with standard error for 22 plant species from a wet meadow community in Central Europe. Asterisks denote statistically significant values of PL in individual plant species.

**Figure 2 plants-09-00640-f002:**
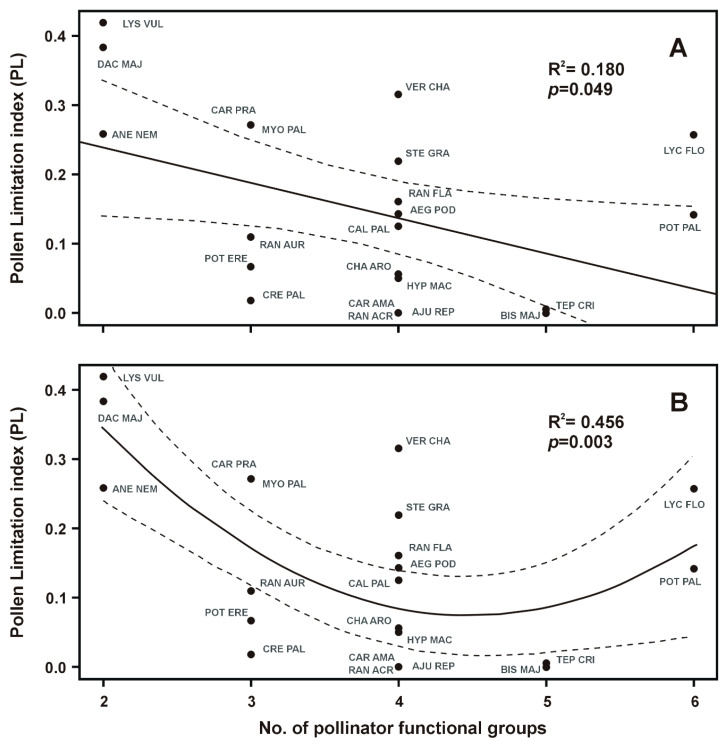
Linear (**A**) and polynomial (**B**) regression of pollen limitation index (PL) with plant specialization (number of functional groups of flower visitors). Dashed lines denote the 95% confidence interval for the model curve. Plant species (see complete list in Table 1) are presented as three letter abbreviations.

**Table 1 plants-09-00640-t001:** Seed production and seed weight for the supplementary hand-pollinated and naturally-pollinated flowers. Asterisks denote p-values of species with statistically significant one-sided test in non-parametric permutation ANOVA. For detail descriptions of the different units used for each species, see section Methods and Materials.

	No. of Seeds						Weight of a Seed (μg)					
Family	Control		Supplemented		PERMANOVA		Control		Supplemented		PERMANOVA	
*Species*	Mean	(std.dev)	Mean	(std.dev)	Pseudo F	*p*	Mean	(std.dev)	Mean	(std.dev)	Pseudo F	*p*
**Apiaceae**												
*Aegopodium podagraria*	16.76	(±8.85)	21.00	(±8.91)	3.75	0.040 *	1.64	(±0.82)	1.95	(±0.90)	1.32	0.138
*Chaerophyllum aromaticum*	20.63	(±11.8)	20.74	(±9.3)	0.00	0.479	2.33	(±1.06)	2.45	(±0.92)	0.75	0.218
**Asteraceae**												
*Crepis paludosa*	30.07	(±13.56)	30.20	(±10.77)	0.00	0.487	0.37	(±0.21)	0.34	(±0.17)	0.15	0.363
*Tephroseris crispa*	55.95	(±30.02)	52.60	(±16.07)	0.24	0.319	0.24	(±0.11)	0.26	(±0.11)	0.44	0.269
**Boraginaceae**												
*Myosotis palustris*	1.25	(±1.22)	1.83	(±1.34)	3.01	0.063	0.23	(±0.20)	0.26	(±0.17)	0.16	0.360
**Brassicaceae**												
*Cardamine amara*	9.72	(±7.05)	7.94	(±7.53)	0.68	0.208	0.05	(±0.03)	0.03	(±0.03)	2.46	0.065
*Cardamine pratensis*	5.89	(±5.2)	8.11	(±5.18)	2.53	0.066	0.15	(±0.09)	0.19	(±0.07)	4.90	0.021 *
**Caryophyllaceae**												
*Lychnis flos-cuculi*	76.32	(±55.12)	100.68	(±46.74)	5.10	0.021 *	0.08	(±0.06)	0.12	(±0.04)	8.43	0.005 *
*Stellaria graminea*	7.54	(±6.00)	9.57	(±4.86)	3.12	0.045 *	0.19	(±0.11)	0.22	(±0.06)	2.99	0.051
**Hypericaceae**												
*Hypericum maculatum*	339.10	(±142.53)	364.45	(±161.06)	1.03	0.165	0.04	(±0.01)	0.04	(±0.01)	0.44	0.259
Lamiaceae												
*Ajuga reptans*	3.40	(±1.12)	3.13	(±1.13)	0.52	0.261	0.99	(±0.38)	1.01	(±0.34)	0.02	0.431
**Orchidaceae**												
*Dactylorhiza majalis* *	0.005	(±0.00)	0.01	(±0.01)	10.15	0.001 *						
**Plantaginaceae**												
*Veronica chamaedrys*	2.00	(±2.95)	3.00	(±3.42)	0.68	0.208	0.08	(±0.10)	0.11	(±0.10)	0.46	0.252
**Polygonaceae**												
*Bistorta major*	79.50	(±45.13)	69.85	(±36.36)	2.16	0.080	5.21	(±1.50)	5.20	(±1.53)	0.00	0.477
**Primulaceae**												
*Lysimachia vulgaris*	12.65	(±16.24)	39.05	(±36.66)	9.29	0.003 *	0.22	(±0.15)	0.24	(±0.13)	0.35	0.281
**Ranunculaceae**												
*Anemone nemorosa*	7.11	(±4.69)	11.61	(±7.01)	10.71	0.002 *	2.13	(±0.99)	2.32	(±0.84)	0.33	0.283
*Caltha palustris*	30.82	(±39.43)	32.82	(±27.1)	0.03	0.429	0.43	(±0.23)	0.47	(±0.20)	0.31	0.308
*Ranunculus acris*	19.33	(±7.08)	15.44	(±8.12)	3.43	0.041	1.23	(±0.23)	1.20	(±0.45)	0.06	0.397
*Ranunculus auricomus*	10.40	(±3.28)	11.80	(±2.91)	3.64	0.041 *	2.01	(±0.48)	2.00	(±0.38)	0.01	0.460
*Ranunculus flammula*	31.89	(±16.94)	37.83	(±12.67)	2.23	0.076	0.34	(±0.15)	0.38	(±0.06)	0.75	0.213
**Rosaceae**												
*Potentilla erecta*	5.55	(±3.14)	6.15	(±3.34)	0.56	0.238	0.54	(±0.24)	0.52	(±0.24)	0.12	0.369
*Potentilla palustris*	219.11	(±80.41)	253.16	(±54.14)	4.05	0.029 *	0.26	(±0.08)	0.28	(±0.11)	0.37	0.276

* In *Dactylorhiza majalis* the capsule weight in grams was used as a proxy for the number of developed seeds.

**Table 2 plants-09-00640-t002:** Linear regressions of selected traits with pollen limitation. Asterisks denote *p*-values of species with statistically significant test.

Traits	F-Statistic	DF	*p*-Value
Specialization (No. of pollinator functional groups)	4.39	20	0.049 *
Clonality	1.24	20	0.278
Dichogamy	0.30	20	0.586
Sugar content	1.29	20	0.268
No. of open flowers	0.66	20	0.426
Self-compatibility	0.70	20	0.411
Autonomous selfing	0.17	20	0.681

**Table 3 plants-09-00640-t003:** The daily nectar sugars production of plant species per flower.

Family	Nectar Sugars (mg)		
*Species*	Flower/Day		std.dev
**Apiaceae**			
*Aegopodium podagraria*	0.0149	±	0.007
*Chaerophyllum aromaticum*	0.0167	±	0.0124
Asteraceae			
*Crepis paludosa*	0.0099	±	0.0123
*Tephroseris crispa*	0.0216	±	0.0269
**Boraginaceae**			
*Myosotis palustris*	0.0039	±	0.0076
**Brassicaceae**			
*Cardamine amara*	0.0184	±	0.0196
*Cardamine pratensis*	0.0193	±	0.0343
**Caryophyllaceae**			
*Lychnis flos-cuculi*	0.2666	±	0.1266
*Stellaria graminea*	0.1185	±	0.0594
**Hypericaceae**			
*Hypericum maculatum*	0.0005	±	0.0003
**Lamiaceae**			
*Ajuga reptans*	0.2150	±	0.0671
**Orchidaceae**			
*Dactylorhiza majalis*	0.0014	±	0.0011
**Plantaginaceae**			
*Veronica chamaedrys*	0.1540	±	0.0548
**Polygonaceae**			
*Bistorta major*	0.0598	±	0.0152
**Primulaceae**			
*Lysimachia vulgaris*	0.0009	±	0.0017
**Ranunculaceae**			
*Anemone nemorosa*	0.0004	±	0.0004
*Caltha palustris*	0.0006	±	0
*Ranunculus acris*	0.0526	±	0.0375
*Ranunculus auricomus*	0.0144	±	0.0074
*Ranunculus flamula*	0.0314	±	0.0175
**Rosaceae**			
*Potentilla erecta*	0.0804	±	0.0722
*Potentilla palustris*	3.2997	±	1.0182

**Table 4 plants-09-00640-t004:** Recorded pollinator spectrum of plant species (total number of visits per 72 h of video recording).

Family	Ants	Beetles	Bumblebees	Butterflies	Honey Bees	Hoverflies	Long-Tonged Flies	Moths	Other Bees	Other Flies	Other Hymenopterans	Total Visits by Plants
*Species*
**Apiaceae**												
*Aegopodium podagraria*	1	49			2	4		2	13	52		123
**Asteraceae**												
*Crepis paludosa*		1		1	25	8			15	1	1	52
*Tephroseris crispa*		3	5	2	1	32			4	19		66
**Boraginaceae**												
*Myosotis palustris*		9				12				29		50
**Brassicaceae**												
*Cardamine amara*		9				12				29		50
*Cardamine pratensis*		2			1	9		1	3	45		61
**Caryophyllaceae**												
*Lychnis flos-cuculi*	3	1		47	76	11		8	4	2		152
*Stellaria graminea*	1	5		1	1	53			5	31		97
**Hypericaceae**												
*Hypericum maculatum*		1	7	2	129	55		1		4		199
**Lamiaceae**												
*Ajuga reptans*	4		22	3		7			1	1		38
Orchidaceae												
*Dactylorhiza majalis*	1	1	1	3	1		1	3				11
**Plantaginaceae**												
*Veronica chamaedrys*		3			1	7			5	11		27
**Polygonaceae**												
*Bistorta major*		5	1	1	21	4			3	7		42
**Primulaceae**												
*Lysimachia vulgaris*		1				11			12			24
**Ranunculaceae**												
*Anemone nemorosa*		2			1	4			1	5		13
*Caltha palustris*	1			1	15	77			3	41		138
*Ranunculus acris*		7	1	1	1	20		1	10	20		61
*Ranunculus auricomus*	1	5				20			1	5		32
*Ranunculus flamula*		16				19			5	16		56
**Rosaceae**												
*Potentilla erecta*					1	94			7	9	2	113
*Potentilla palustris*		2	9	7	3	52		1	9	18		101
**Total visits by groups**	12	122	46	69	279	511	1	17	101	345	3

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
