# Peer review of "Are Reproductive Traits Related to Pollen Limitation in Plants? A Case Study from a Central European Meadow"

_plants, 2020, doi:10.3390/plants9050640_

Round 1

Reviewer 1 Report

I really liked this paper. It is straight forward the science is good, and they give a sober and even handed discussion of the results, their conclusion, and the significance of the subject. in many cases bringing up important points sometimes over looked by other investigators. They state clearly when their data did not conform to what was in the literature.

The conservative interpretation I always consider good science. However, the regression shown in Figure two (2) are interesting that both are significant and each one can lead to different interesting conclusions. The authors explain this situation in lines 123-124 in an acceptable way. It might be a bit indulgent, but also inspiring to other investors if the authors spoke briefly about what those different conclusion might be depending on verification of the model in 2A as opposed to 2B. This is not necessary but I think some speculation might be warranted.   

Please be sure to either italicize scientific names or put them in bold.

Author Response

Reviewer 1

Comments and Suggestions for Authors

I really liked this paper. It is straight forward the science is good, and they give a sober and even handed discussion of the results, their conclusion, and the significance of the subject. in many cases bringing up important points sometimes over looked by other investigators. They state clearly when their data did not conform to what was in the literature.

The conservative interpretation I always consider good science. However, the regression shown in Figure two (2) are interesting that both are significant and each one can lead to different interesting conclusions. The authors explain this situation in lines 123-124 in an acceptable way. It might be a bit indulgent, but also inspiring to other investors if the authors spoke briefly about what those different conclusion might be depending on verification of the model in 2A as opposed to 2B. This is not necessary but I think some speculation might be warranted.   

* We extended the discussion by adding a paragraph where we briefly mentioned  possible differences in models: However, assuming the validity of the linear model, we expect higher diversity in conspecific pollen load with increasing number of pollinator functional groups because of pollen grains coming from a wider range of donors. It could stimulate pollen competition and successful pollination. On the other hand, assuming the validity of the unimodal model which has much higher significance value than the linear model, increasing the number of pollinator functional groups may involve less-specific pollinators. These may clog stigmas with higher loads of heterospecific pollen, which could decrease the reproductive success of plants. Nevertheless, we can only speculate about the accuracy of using one model, because a detailed study on pollinator group effectiveness for each particular plant species would be necessary to make an unambiguous conclusion.”

Please be sure to either italicize scientific names or put them in bold.

* We corrected the formatting of scientific names in lines 97-101.

Reviewer 2 Report

This is an interesting paper which show clearly pollen limitation in the tested habitat, with increased global concern regarding the availability of pollinators in natural and managed systems, the findings are important. However, much information is missing (detailed description of the experimental site, how exactly the hand pollination performed) description of the pollinators and amount of sugar produced in different flowers, please see below).

Line 64; "Although self-incompatible plants are generally expected to be more pollen limited than self-compatible plants, this assumption may not always be true [17]"- This is a very strange statement, please elaborate, how it can be, as self pollen, by definition, have much higher chance to arrive to the stigma.

Table 1; number of seed per what ? per fruit ? please indicate the unit(s) you count

Table 1; Dactylorhiza majalis, have exactly the same number of seeds (0.01) for both the treatment and  the control, so how it can be that they are statistically different?  

Line 87; you focused on nectar reward, but many pollinators attract to the flower to consume pollen.. can you refer to this as well?

Line 238; I presume you focused only on insect-pollinated plants ?

Please provide more information on the study site; climate? soil ? nearby urban/ agricultural landscape ? any use of agrochemicals in the area ?

Line 243 individual plants, please explain how you have performed the hand pollination

Line 276- 281; please add those data to your paper (the amount of sugar produced in flowers of the different species)

Lines 282-290; please add  a table with that information (which pollinators visit which plants) to the ms.  

Discussion

- Please discuss which were the main pollinators in your habitat and explain why they were dominant in that habitat.

-Please refer to the fact that you have results from only one season from limited time and that under different climatic conditions, the results might be different

-Your discussion is very general; please refer to your results and explain situations with high PL values (for example, see in the literature which are its main pollinator, and maybe they are absent in your habitat) and those with low PL values.

Author Response

Reviewer 2

Comments and Suggestions for Authors

This is an interesting paper which show clearly pollen limitation in the tested habitat, with increased global concern regarding the availability of pollinators in natural and managed systems, the findings are important. However, much information is missing (detailed description of the experimental site, how exactly the hand pollination performed) description of the pollinators and amount of sugar produced in different flowers, please see below).

Line 64; "Although self-incompatible plants are generally expected to be more pollen limited than self-compatible plants, this assumption may not always be true [17]"- This is a very strange statement, please elaborate, how it can be, as self pollen, by definition, have much higher chance to arrive to the stigma.

*  We added an explanation after the statement:  ….“As discussed by García-Camacho and Totland [17], self-compatible species may potentially receive more compatible pollen on their stigmas than self-incompatible species, but resource limitation might not let them take advantage of it. Thus, constraints from specific abiotic conditions could theoretically explain the similarity between self-compatible and self-incompatible species [17].“

Table 1; number of seed per what ? per fruit ? please indicate the unit(s) you count

* We used different units for different species and we described it in detail in the section Materials and Methods: Pollen limitation. We refer to this information in the Table 1 caption. We are afraid that adding this information into Table 1 could make the table less legible.

Table 1; Dactylorhiza majalis, have exactly the same number of seeds (0.01) for both the treatment and  the control, so how it can be that they are statistically different?  

* We corrected the values for the number of seeds, which were affected by rounding to two decimal places. Now the readers can see the differences. Because Dactylorhiza majalis produces numerous tiny seeds, we used the capsule weight as a proxy for the number of developed seeds.

Line 87; you focused on nectar reward, but many pollinators attract to the flower to consume pollen.. can you refer to this as well?

* We are aware of the importance of pollen offer for plant attractivity for visitors, but unfortunately we do not have adequate data about pollen production for the investigated plant species.  We added this information into the following sentence in the Discussion: “...But nectar is not the only reward which is offered by plants. Pollen is also a very important attractant for visitors. Unfortunately, we do not have adequate data about pollen production for the investigated plant species.“

Line 238; I presume you focused only on insect-pollinated plants ?

* Yes, we focused only on insect-pollinated plants. We added this information into the following sentence in the Methods: We collected data on the 22 most abundant insect-pollinated plant species flowering between the beginning of May and mid-July 2017”.

Please provide more information on the study site; climate? soil ? nearby urban/ agricultural landscape ? any use of agrochemicals in the area ?

* We provided more information on the study site. We added these sentences: “…Mean annual temperature is 6.4 °C and annual precipitation is 745 mm. The subsoil is formed by fluvial sandy loam and sandy gravels. The meadow, with an area of 1.2 ha, is isolated from the surrounding agricultural landscape by a high forest. The meadow is mowed once a year. By phytosociological classification [39], the meadow belongs to the alliance Calthion palustris with vegetation dominated by Agrostis canina, Scirpus sylvaticus, Lysimachia vulgaris, and Filipendula ulmaria…."

Line 243 individual plants, please explain how you have performed the hand pollination

* We added the following information into the Methods: “…We collected anthers with visible pollen grains (verified by hand lens) and rubbed them over the receptive stigmas (successful deposition was again verified by hand lens). For compact inflorescences, capitulas, or umbels, we rubbed the whole donor unit over the recipient one. ….”

Line 276- 281; please add those data to your paper (the amount of sugar produced in flowers of the different species)

* We added information about the daily nectar sugars production of each plant species per flower to Table 3.

Lines 282-290; please add  a table with that information (which pollinators visit which plants) to the ms.  

* We added the recorded pollinator spectrum of plant species to Table 4.

Discussion

- Please discuss which were the main pollinators in your habitat and explain why they were dominant in that habitat.

* We extended the discussion with the following sentences: …“However, the main pollinator group recorded on the studied locality appears to be hoverflies, feeding on both nectar and pollen. A wide range of hoverfly larvae are associated with accumulations of wet, rotting vegetation in ponds and ditches, which are common nearby. The other abundant groups were other flies and honey bees (Table 4).“

-Please refer to the fact that you have results from only one season from limited time and that under different climatic conditions, the results might be different

* Now, we refer to these limitations at the end of the discussion: “Our results describe pollen limitation based on observation from a single season. As seasonal course of climate affects both plant phenology and insect activity, thus we cannot exclude that pollen limitation will be different in other seasons.”

-Your discussion is very general; please refer to your results and explain situations with high PL values (for example, see in the literature which are its main pollinator, and maybe they are absent in your habitat) and those with low PL values.

* We are newly discussing this question as following:” Species with high PL values have specific ecological features. Lysimachia vulgaris is pollinated by highly specialized oil-collecting Macropis bees [34]. The species is also dominant in the locality, producing many flowers at same time, which suffer from competition for pollinators. An orchid Dactylorhiza majalis offers no reward to pollinators, which are deceived by showy flowers. Deceptive orchids typically produce little fruits [35]. Anemone nemorosa is flowering very early in the vegetation season when visitors are limited by unexpected weather conditions, especially by low temperature [36]. Species with low PL mainly belong to a group of plants with many, generalized flowers with easily available nectar rewards.

Round 2

Reviewer 2 Report

The authors addressed very well the comments they received, I recommend accepting the paper